# A Hybrid Fuzzy MCDM Methodology for Optimal Structural System Selection Compatible with Sustainable Materials in Mass-Housing Projects

Ebrahim Aghazadeh *, Hasan Yildirim and Murat Kuruoglu

Department of Civil Engineering, Istanbul Technical University, 34469 İstanbul, Turkey
* Correspondence: aghazadehe@itu.edu.tr; Tel.: +98-910-1415828

**Abstract:** The present paper aimed to propose a new support approach to choosing the optimal structural system in accordance with sustainable materials in mass-housing projects. To this end, an integrated fuzzy multi-criteria-decision-making (fuzzy MCDM) method was used to identify the criteria affecting sustainable material selection. The proposed approach consists of a three-phase protocol: In phase I, the literature was used to create a database encompassing 42 factors affecting the selection of materials. These factors were classified as four indicators (economic, environmental, socio-cultural, technical-executive) in accordance with the sustainable development aspects. In phase II, the fuzzy Delphi method (FDM) was used to screen the key factors. In phase III, an integrated fuzzy SWARA–ARAS method was used to prioritize the optimal structural system for a case project: evidence from Iran. The results of selecting the structural systems based on 14 efficient key factors showed that the Light Steel Frame (LSF), Insulation Concrete framework (ICF), and the Prefabricated Reinforced Concrete System (PRC) systems have the highest priority to achieve the goals of sustainable material selection, respectively.

**Keywords:** sustainable materials; optimal structural systems; integrated fuzzy MCDM approach; mass-housing projects

## 1. Introduction

The evolution of construction operations and the advancement of materials technology have devastating effects on the environment and pose a serious threat to humans and other organisms [1,2]. Conscious selection of materials, correct consumption, and material recycling and reuse can effectively contribute to environmental protection and the mitigation of their negative effects. Therefore, knowledge of useful and effective decision-making methods in material selection is of vital importance [3,4].

Materials determine the nature of a building [5]. Therefore, material selection has always been known as one of the key factors contributing to achieving sustainable development goals in the construction industry [6]. How the material selection can help achieve sustainable development goals and what key indicators affect this selection are questions that can be answered in different ways, depending on the goals and needs of a project [1,7,8]. Decision making for selecting materials that can contribute to achieving sustainable development goals in the construction industry depends on various indicators. Identification and evaluation of these indicators not only helps with making conscious material choices, but also helps with improving the quality of buildings, increasing their service life, and even guaranteeing their optimal operation [9]. This process follows an interactive procedure that requires the compatibility of selected materials with the main aspects of sustainable development (economic, social, and environmental) [4,7,10]. Fulfillment of this interactive process is very complicated due to the mainly high uncertainties associated with the decision–making process. Moreover, in some cases, this selection can be very controversial [1]. This is of vital importance in construction projects, especially in

mass-housing projects that require large volumes of consumer products, and the slightest negligence in this field can incur additional costs and lead to a waste of time and distraction from the sustainable development goals in the construction industry [3].

Sustainable development actually seeks to propose architectural solutions for the well-being and coexistence of solid elements, organisms, and humans [8,9]. In other words, in sustainable design, efforts should be made to deal with the needs of building residents, improve the quality of human life, and reduce adverse effects on the natural environment at the same time. During their lifetime, buildings impact the local and global environment through various human activities and inter-related mechanical processes [7]. This change starts from the regional microclimate and finally triggers transformations in the global ecosystem. Thus, it can be argued that consideration of different aspects of sustainability in the optimal material selection and application is one of the most important things in sustainable design [2,11–14].

Material selection planning coupled with consideration of the different dimensions of sustainable development can lead to a breakthrough in the construction industry [15]. This planning provides a broad outlook on the social, economic, and environmental impacts of material production or use in the construction industry [13–17]. Material selection planning is feasible only when a decision–making support system for the selection of sustainable materials in the construction industry is developed. This system is affected by many factors and insufficient information and description of them based on different semantic variables related to those factors will give rise to uncertainties in the results [8]. In addition, since expert opinions based on verbal expressions are involved in the development of these systems, their results are mostly ambiguous with faulty mentalities [18–24]. The integrated fuzzy multi-criteria-decision-making approaches are able to solve these ambiguities. In addition, by taking into account the effect of uncertainties in the decision–making process, these approaches can serve as a perfect tool for the collection of accurate information [16,17].

The present study aimed to propose a decision-making support system for evaluating the factors affecting sustainable material selection in the construction industry. Considering the extensive uncertainties associated with the decision-making process for selecting the sustainable material in mass-housing projects, a combination of different fuzzy multi-criteria-decision-making approaches was utilized to create a local support system. Therefore, first, the most important factors affecting the material selection were identified through the literature studies and collection of experts' opinions. Then, the identified factors were screened, and key factors were extracted using statistical methods. At the next step, fuzzy multi-criteria-decision-making approaches were integrated to prioritize the screened factors and to select a sustainable structural system for mass-housing projects. The proposed support system was implemented in order to apply a realistic scenario to select an optimal structural system for a mass-housing project in a residential area in Iran. In this scenario, an executive company needed to make a decision to choose building materials in accordance with the principles of sustainable development in a mass-housing project. This company was required to consider all possible important criteria that determine the sustainability in buildings from the point of choosing the type of proposed structural system. Although the issue of sustainable materials selection in construction projects has been investigated in many studies in the past, the main difference and innovation of our research compared to others is that it considered a large number of criteria for selecting sustainable materials on the one hand. On the other hand, presenting a hybrid support model based on MCDM methods in a fuzzy environment to implement these criteria for choosing an optimal structural system in mass-housing projects is a significant innovation.

## 2. Review of the Literature

Identification and prioritization of the key factors affecting the selection of sustainable materials in the construction industry can increase the ability of famous construction triangle sides (cost, time and quality) to develop a decision-making support system [14,18,25]. The available literature could provide effective results concerning the development of

decision-making support systems in the material selection process [17,20,22,26–31]. Despite the fact that extensive studies have been carried out in this respect (some of which are listed in Table 1), some gaps have made it necessary to conduct further studies on the identification and prioritization of the factors affecting the development of decision-making support systems in the process of sustainable material selection and employing coherent methods. These include methods such as the new hybrid fuzzy multi-criteria-decision-making approaches in the prioritization process in order to fulfill the different dimensions of sustainable development. On the other hand, the relevant literature indicates a lack of decision-making information required for the prioritization of materials in the presence of uncertainties. These shortcomings, coupled with the nonintegrated literature which is not capable of identifying the uncertainties involved in the development of a decision-making support system, have given the present researchers sufficient motivation to conduct the current study.

**Table 1.** Quick summary of the literature review.

| Authors (Ref.) | Methodologies | Goal |
| --- | --- | --- |
| Rao and Davim, 2008 [18] | AHP and TOPSIS (Technique for order performance by similarity to ideal solution) | Material selection for a given engineering design |
| Zhou et al., 2009 [16] | Artificial Neural Networks (ANN) and Genetic Algorithms (GA) | A decision support optimization system for sustainable material selection |
| Chatterjee et al., 2009 [17] | VIKOR, ELECTRE | Selection of materials |
| Onut et al., 2009 [11] | Fuzzy ANP (Analytic Network Process) and TOPSIS | Selection of the appropriate material handling equipment |
| Tuzkaya et al., 2010 [13] | Fuzzy ANP and Fuzzy PROMETHEE | Selection of material handling equipment |
| Akadiri et al., 2012 [6] | FEAHP | A decision-making model for building material selection |
| Bakhoum and Brown, 2012 [3] | – | A sustainable scoring system for materials |
| Rahman et al., 2012 [19] | TOPSIS | A decision support system to select the optimal roofing materials |
| Liu et al., 2014 [20] | DANP and VIKOR | Material selection with target-based criteria |
| Zhao et al., 2016 [21] | GRA | Commercially available materials selection in sustainable design |
| Govindan et al., 2016 [14] | hybrid MCDM method | Proposed a model to select sustainable material |
| Gul et al., 2018 [23] | presented a fuzzy logic-based PROMETHEE | Select the material for an automotive instrument panel |
| Khoshnava et al., 2018 [24] | Hybrid MCDM method | Ranking the green building material criteria based on sustainability |
| Kiani et al., 2018 [25] | VIKOR | Select the material for repair structural concrete |
| Mousavi-Nasab and Sotoudeh-Anvari, 2018 [26] | COPRAS (Complex Proportional Assessment), VIKOR and TOPSIS | Suggestion a new MCDM-based model for sustainable material selection |
| Mahmoudkelaye et al., 2019 [8] | ANP | Proposed a ranking model for sustainable material selection |
| Chen et al., 2019 [27] | QFD (Quality Function Deployment) and ELECTRE | Sustainable building material selection |
| Singh et al., 2020 [4] | Fuzzy AHP and M-TOPSIS | Choose the composite material based on mechanical and structural applications |
| Rajeshkumar et al., 2020 [28] | Structural questionnaire survey | Material selection in high rise buildings |
| Emovon and Oghenenyerovwho, 2020 [7] | A systematic review | Application of MCDM methods in material selection |

**Table 1.** *Cont.*

| Authors (Ref.) | Methodologies | Goal |
|---|---|---|
| Mayhoub et al., 2021 [29] | AHP | Achieving the sustainable building façades |
| Agrawal, 2021 [5] | SAW (Simple Additive Weighting), MOORA and TOPSIS | Sustainable material selection for additives manufacturing technologies |
| Chen et al., 2021 [30] | QFD and TOPSIS | Sustainable building material selection |
| Majer et al., 2022 [31] | WSM-weighted sum method | Selection of external walls based on user priority |
| Sahlol et al., 2021 [2] | System dynamics and AHP | Sustainable building materials assessment |

## 3. Materials and Methods

The methodology in the present study can be classified according to two main factors: the research objectives and the data collection method. In other words, the present study falls within the category of applied developmental studies in terms of objectives, which seek to develop a decision-making support tool to evaluate the effective criteria involved in sustainable material selection in construction projects. This research falls within the category of descriptive field studies (in terms of data collection method) being conducted through the objective inspection and examination of material selection procedures in construction projects carried out by contracting companies in Iran. The hybrid approach used in the present study falls within the category of heuristic studies. In heuristic integrated design, first, the qualitative data, and then, the quantitative data are collected. Qualitative data are collected to scrutinize the phenomenon under study. Then, the researcher draws on the findings derived from the qualitative data to collect the quantitative data, and in this way, could generalize the findings [32,33]. The difference between this method and other methods in the present article is the lack of consensus among researchers over selection of sustainable materials for construction projects in Iran.

### 3.1. Instruments and Techniques

3.1.1. Sample of the Study

The target population of the study includes a 12-member team of civil engineers and senior project managers with specialties and skills in different fields such as construction materials and sustainable development in the construction industry. Experts were selected using the snowball sampling method based on the researcher's personal experience. This type of sampling is often used when access to all members of the population is difficult to obtain or the population size is limited. Among the advantages of this method, one could mention simplicity and no need for extensive planning and physical work compared to other sampling methods [34]. These experts were selected considering the following four main characteristics: 1. Knowledge and experience in the field under discussion. 2. Willingness to participate in the discussion. 3. Sufficient time to participate in the survey. 4. Effective communication skills. The demographic information of the selected individuals as a panel of experts is given in Table 2.

**Table 2.** Detailed information of experts.

| Parameters | Component | Frequency | Frequency Percentage |
|---|---|---|---|
| Work position | Project Chief Supervising Engineer | 3 | 25 |
| | Site Manager | 7 | 58.33 |
| | Senior Project Manager | 2 | 16.67 |
| Experience in the sustainable construction field | Between 5 and 10 years | 5 | 41.67 |
| | Between 10 and 20 years | 5 | 41.67 |
| | More than 20 years | 2 | 16.66 |

**Table 2.** *Cont.*

| Parameters | Component | Frequency | Frequency Percentage |
|---|---|---|---|
| Field of work | Employer | 4 | 33.33 |
| | Contractor | 3 | 25.00 |
| | Consultant | 5 | 41.67 |
| Skill and expertise related to sustainable construction | Number of workshops and training seminars related to materials | At least 22 | |
| | Number of participations in sustainable construction projects | At least 19 | |
| | Number of participations in mass-housing projects | At least 21 | |

### 3.1.2. Questionnaire and Validation

Experts use their mental abilities to provide feedback in a poll program. Human judgment is generally associated with a degree of uncertainty because the human mind does not fully quantify opinions. In the presence of data uncertainty, fuzzy approaches must be used to ensure the results' accuracy to make real-world decisions [35]. The nature of fuzzy logic in such a situation is decision-making based on existing relative differences between the effects of opinions in achieving the answer, and ambiguity in some qualitative and quantitative views is included in the answers [36]. The impact of the decision makers' judgments can be reduced using this approach, which generally limits fuzzy events and objects in unknown circumstances by using inaccurate and nonquantitative words [37].

Zadeh [38] developed the theory of fuzzy sets to deal with the uncertainties caused by inaccuracies and ambiguities in decision making. In uncertain situations, this theory can mathematically express many ambiguous concepts, variables, and systems. In uncertain conditions, this theory provides the foundation for reasoning, inference, control, and decision making [39]. A fuzzy set is a collection of objects with a continuous degree of membership. A membership function, which assigns each object a membership rating between 0 and 1, characterizes such a set [40]. In fuzzy sets, the definition of a membership function is context-dependent. A fuzzy number, according to this theory, is a specific fuzzy set defined by Equation (1), in which $x$ accepts the real values of the members of the set $R$ with a membership function $\mu_{\widetilde{A}}(x)$.

$$\widetilde{A} = x \in R / \mu_{\widetilde{A}}(x) \tag{1}$$

Several standard membership functions have been introduced in the literature on fuzzy set theory. One of the most widely used is the triangular membership function [41]. A triangular fuzzy number (TFN), $A$ $(a^l, a^m, a^u)$, is a number with the membership function of linear fractions A (Equation (2)), as shown in Figure 1.

$$\mu_x(x) = \begin{cases} (x - a^l)/(a^m - a^l) & a^l \leq x < a^m \\ 1 & x = a^m \\ (a^u - x)/(a^u - a^m) & a^m < x \leq a^u \\ 0 & otherwise \end{cases} \tag{2}$$

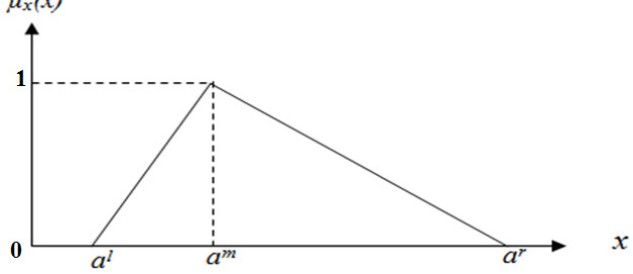

**Figure 1.** Triangular and fuzzy membership functions (adapted with permission from [41]).

The parameters $a^l$, $a^m$, and $a^u$, respectively, represent the lowest, most likely (most promising), and highest values of a possible value that describes a fuzzy event. When $a^l = a^m = a^u$, then $A$ is a nonfuzzy number, according to the contract. References can be found for algebraic operations on two fuzzy triangular numbers [40]. It is easy to express the value of an object using fuzzy numbers to express the decision makers' qualitative evaluations. As a result, the use of fuzzy numbers in decision-making methods has grown in popularity [42].

The answers to the questionnaires were divided into three categories in the decision-making problem of the present study: closed-ended, fuzzy-form, and open-ended. The present study used the five-point Likert scale to measure the identified critical factors in the first section questionnaires. The importance of the items was determined using a range of ambiguous linguistic expressions of very low, low, medium, high, and very high importance, the details of which are explained below. The degrees of preference for efficient critical factors were compared to each other using verbal expressions in the second part of the questionnaires. The third section of the questionnaires compared the current states of the selected projects in terms of final, efficient, critical factors.

The main criteria were used to determine the validity and reliability of the items identified in the questionnaires. The term "validity" refers to the goal that the test is supposed to achieve. The validity of a questionnaire is based on two principles: clarity and simplicity [32]. This study used the content validity ratio (*CVR*) indicator developed by Lawshe [43] (based on Equation (3)) to determine the validity of the questionnaires.

$$CVR = \frac{n_e - n/2}{n/2} \tag{3}$$

The total number of specialists is denoted by $n$, and $n_e$ denotes the number of specialists chosen as the necessary option. The number of specialists at this phase is 12, and the acceptable validity is equal to 0.57, as determined by the minimum *CVR* and scoring specialists [44].

Coder, parallel, bisection, test-retest, and Cronbach's alpha are methods for determining the reliability of any questionnaire [45]. This study used the Cronbach's alpha method and determined the reliability coefficient $R_\alpha$ (according to Equation (4)) for the main criteria after collecting the results of the first part of the questionnaires.

$$R_\alpha = (k/k - 1)\left(1 - \sum \sigma_j^2/\sigma^2\right) \tag{4}$$

where $k$ represents the total number of test questions. $\sigma_j^2$ denotes variance of the $j$-th question scores, and $\sigma^2$ denotes variance of the total questions scores. The Cronbach's alpha coefficient must be, at minimum, equal to 0.7 to be considered reliable [44].

### 3.2. Problem-Solving Process

According to the flowchart shown in Figure 2, the problem-solving process in the current study, which is developing a model for the selection of sustainable materials in mass-housing projects, necessitates maintenance. The maintenance process consists of three main parts and a three-phase protocol. In continuation, each of these phases is explained with more details.

### 3.2.1. Phase I: Preparing a Database

In phase I, a database encompassing factors affecting the selection of sustainable materials in construction projects was developed and the optimal construction systems are identified. This was performed to help the contracting companies achieve sustainability in Iranian construction projects. Literature findings, intuitions, as well as expert experiences and judgments are among the most commonly used techniques for identifying and differentiating the factors. Therefore, in the present study, the foreign and domestic literature, global information network, and sustainable material regulations and guidelines were

reviewed, and a field survey was conducted. Finally, communication, interviews, questionnaire analysis, and inquiries made by experts were used to identify the effective factors.

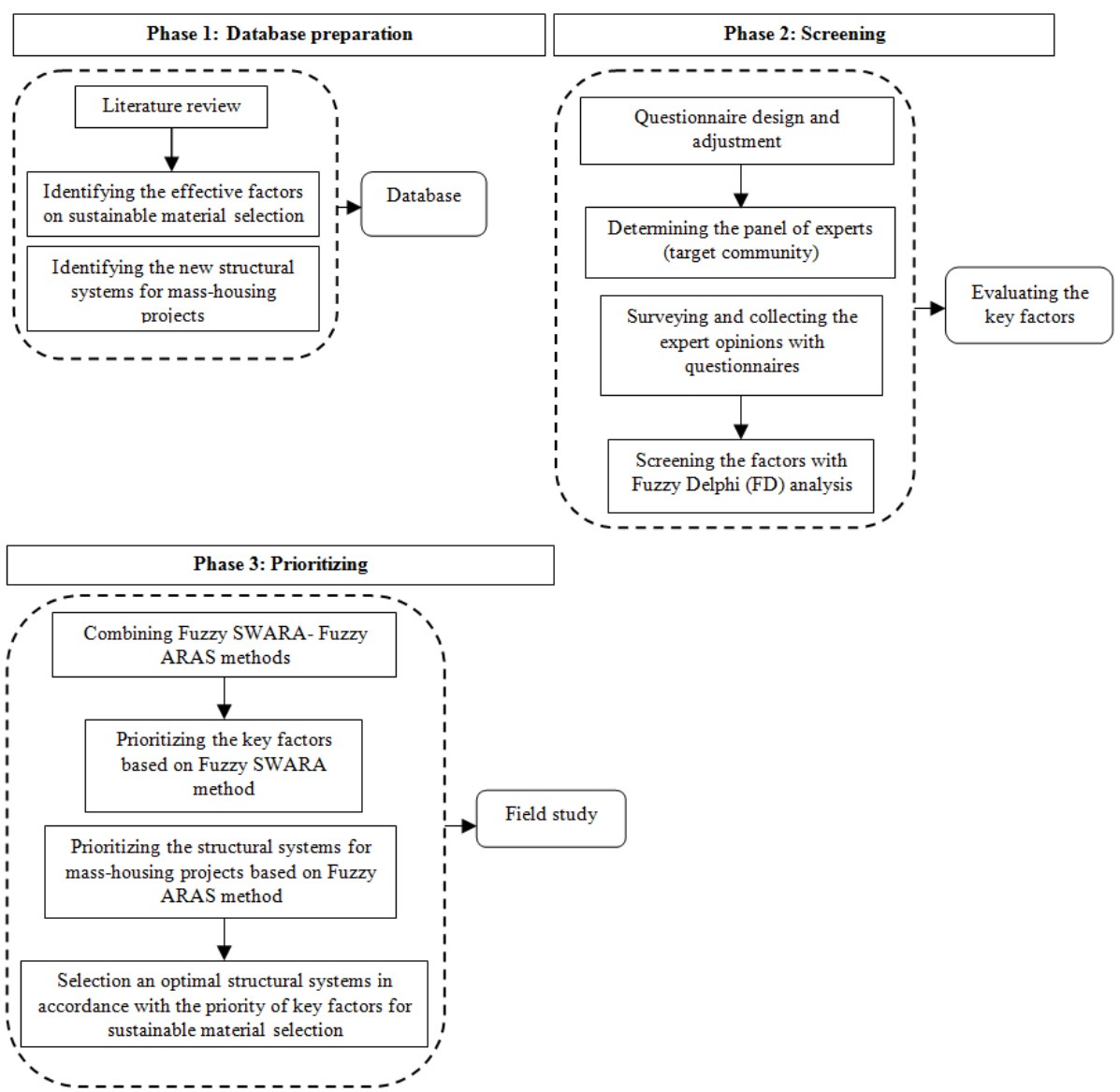

**Figure 2.** Flowchart of problem-solving process.

As mentioned before, the present study is an executive attempt to develop a decision-making support model for selecting the optimal structural system. Next, the selected structural system was utilized to execute the structural frames used in mass-housing projects, while taking into account compatibility with the sustainable material selection procedures. Several structural systems with specific advantages and disadvantages were proposed for this purpose. According to the initial opinion of the employer of a mass construction project, five types of new structural systems known as the Light Steel Frame (LSF), Prefabricated Reinforced Concrete System (PRC), Insulation Concrete framework (ICF), 3D Sandwich Panels (3DP), and the Tronco System (TRC) were proposed to the project contractor for constructing the building. The basic specifications and the executive form of the proposed structural systems are presented in Figure 3 and Table 3, respectively.

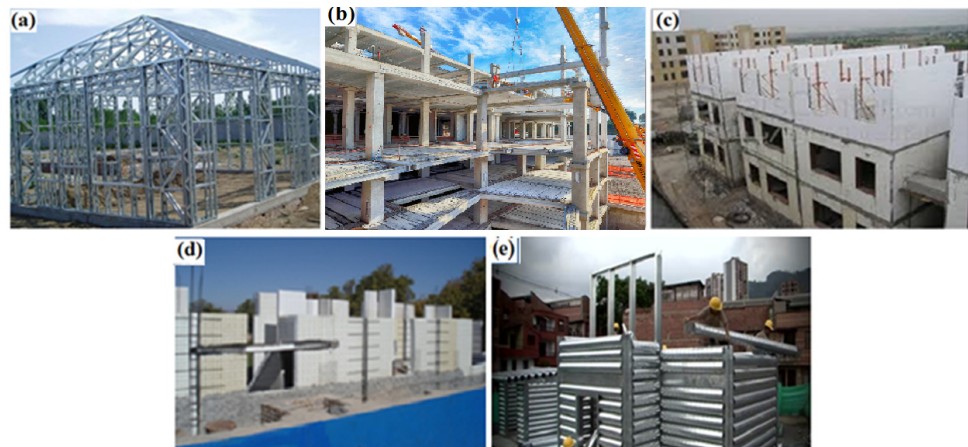

**Figure 3.** Proposed options for building skeletons in mass construction projects, (**a**) LSF, (**b**) PRC, (**c**) ICF, (**d**) 3DP, and (**e**) TRC.

**Table 3.** Basic information of the structural systems.

| Structural Systems | Information |
| --- | --- |
| Light Steel Frame (LSF) | LSFs are used to construct buildings with a limited number of floors (usually up to 5 floors). LSF is composed of cold-rolled steel sheets to provide stability. The foundation thickness is very small in this system due to the small loads applied on the building; the system is highly resistance to earthquakes due to its light weight without need for traditional and heavy materials. Moreover, due to the uniform distribution of forces throughout the building, the system is recognized as a highly safe system. Environmental friendliness, flexibility, high durability, dimensional stability, and stiff sections are the main advantages of this system. |
| Prefabricated Reinforced Concrete (PRC) | In this system, the concrete parts that are prefabricated according to the maps are transported from the factory to the construction site. Since concrete parts are prefabricated, there are no considerable dimensional and proportional limitations in the architectural design of this system. The system is characterized by its fast and easy implementation and, thus, short duration between investment and operation. In addition, short execution time and cost as well as high service life are among the advantages of this system. |
| Insulating Concrete Framework (ICF) | This system consists of reinforced concrete as the load-bearing component and expanded polystyrene (EPS) panels as concrete formwork and thermal insulators. Earthquake resistance, acoustic and thermal insulation, low construction costs, lack of architectural form limitations, long durability and service life, easy operation of the building, integration into other systems, and fast execution, the possibility of implementation in different seasons of the year, formwork independence, allowing for embedding ducts and pipes in the walls, and easy transportation are among the advantages of this system. |
| 3D Sandwich Panels (3DP) | The 3DP is a suitable and effective system that can significantly simplify the building construction process. The system is characterized by lightness, strength, integrity, insulation, and fast and easy implementation, which make the system fully compliant with safety and other relevant standards. The advantages of this structural system include reduced weight, structural rigidity and limited displacement, decreased prime cost, reduced spaces occupied by walls, rapid implementation, and easy installation of electrical and mechanical ducts, etc. |
| Tronco System (TRC) | TRC consists of simple frames with cold-rolled metal elements. This system is usually applicable in low-rise buildings. During operation and implementation of this system, the door frames, windows, and electrical and mechanical elements are installed within predetermined spaces. Since the empty spaces within pipes, walls, and ceiling are filled by EPS panels, this system is the best choice in terms of energy saving and light weight. Considering the advantages, this system resembles the LSF system but is constructed and implemented in a different and more expensive way. |

3.2.2. Phase II: Screening of Factors Identified by Fuzzy Delphi Method (FDM)

The factors were refined and screened using the fuzzy Delphi method (FDM), and considering the opinions of the 12-member experts in phase II, after identifying and classifying

them in phase I. The aim of this phase was screening and finalizing the most important factors. The Delphi technique is an efficient method in terms of utilizing a questionnaire to collect the experts' opinions, insights, experiences, and perceptions [46]. In most cases, the Delphi method is used in multiple rounds. The collected data are analyzed at the end of each round. The new data and questions are presented to the experts in the next round [47]. The FDM was used in three rounds in the current study. During the first and second rounds of removing and merging factors with repetitive content and conceptual overlap from the initial list, some new and remaining factors were added, based on the experts' opinions. The invoices were finally sorted and separated in the second round after they had received final approval. Based on the experts' opinions and verbal expressions of the Likert spectrum, the importance of the final factors was determined in the third round, when the first questionnaire was distributed. The verbal expressions were fuzzified to determine the importance of the factors using the triangular fuzzy numbers (TFNs) shown in Figure 4 and Table 4.

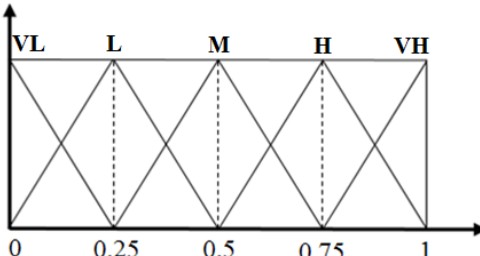

**Figure 4.** Triangular fuzzy membership function (Reprinted with permission from [41]).

**Table 4.** Triangular fuzzy numbers based on 5-point Likert spectrum [41].

| Significance (Verbal Phrase) | Triangular Fuzzy Number | Fuzzy Value |
| --- | --- | --- |
| Very Low (VL) | $\widetilde{1}$ | (0, 0, 0.25) |
| Low (L) | $\widetilde{2}$ | (0, 0.25, 0.5) |
| Medium (M) | $\widetilde{3}$ | (0.25, 0.5, 0.75) |
| High (H) | $\widetilde{4}$ | (0.5, 0.75, 1) |
| Very High (VH) | $\widetilde{5}$ | (0.75, 1, 1) |

After collecting the experts' opinions from n questionnaires, the fuzzy average of opinions for each factor (Equation (5)) was calculated. Then, a crisp number for the importance of each of the critical factors was determined using the center of gravity method (Equation (6)) for defuzzification.

$$\widetilde{\overline{A}} = \left(\overline{a}^l, \overline{a}^m, \overline{a}^u\right) = \left(\frac{\sum_{i=1}^{n} a^l}{n} + \frac{\sum_{i=1}^{n} a^m}{n} + \frac{\sum_{i=1}^{n} a^u}{n}\right) \tag{5}$$

$$\left.\begin{array}{l} x_{max}^l = \left(a^l + a^m + a^u\right)/3 \\ x_{max}^m = \left(a^l + 4a^m + a^u\right)/6 \\ x_{max}^u = \left(a^l + 2a^m + a^u\right)/4 \end{array}\right\} \xrightarrow{\text{Crisp number}} \max\left(x_{max}^l, x_{max}^m, x_{max}^u\right) \tag{6}$$

Finally, any factor with a crisp value greater than 0.5 was considered an efficient critical factor, while factors with a crisp value less than 0.5 were counted as inefficient.

### 3.2.3. Phase III: Prioritization with Hybrid Fuzzy Method (Fuzzy SWARA–Fuzzy ARAS)

Due to the large number of factors affecting the selection of sustainable materials (which were extracted in the previous phase), the possibility of errors in the quantitative

analyses related to evaluating these factors and the studied building systems were very high. Therefore, using hybrid fuzzy multi-criteria-decision-making methods to develop a decision support system would lead to good results. This phase identified and prioritized the more significant factors in the selection of sustainable materials, on the one hand, and the selection of preferred building systems for implementation of mass-housing projects, on the other hand. The used hybrid multi-criteria-decision-making method was based on fuzzy SWARA (Fuzzy Stepwise Weight Assessment Ratio Analysis) and fuzzy ARAS (Fuzzy Additive Ratio Assessment) methods. The flowchart shown in Figure 5 summarizes this process. The details of these methods are introduced in the next section.

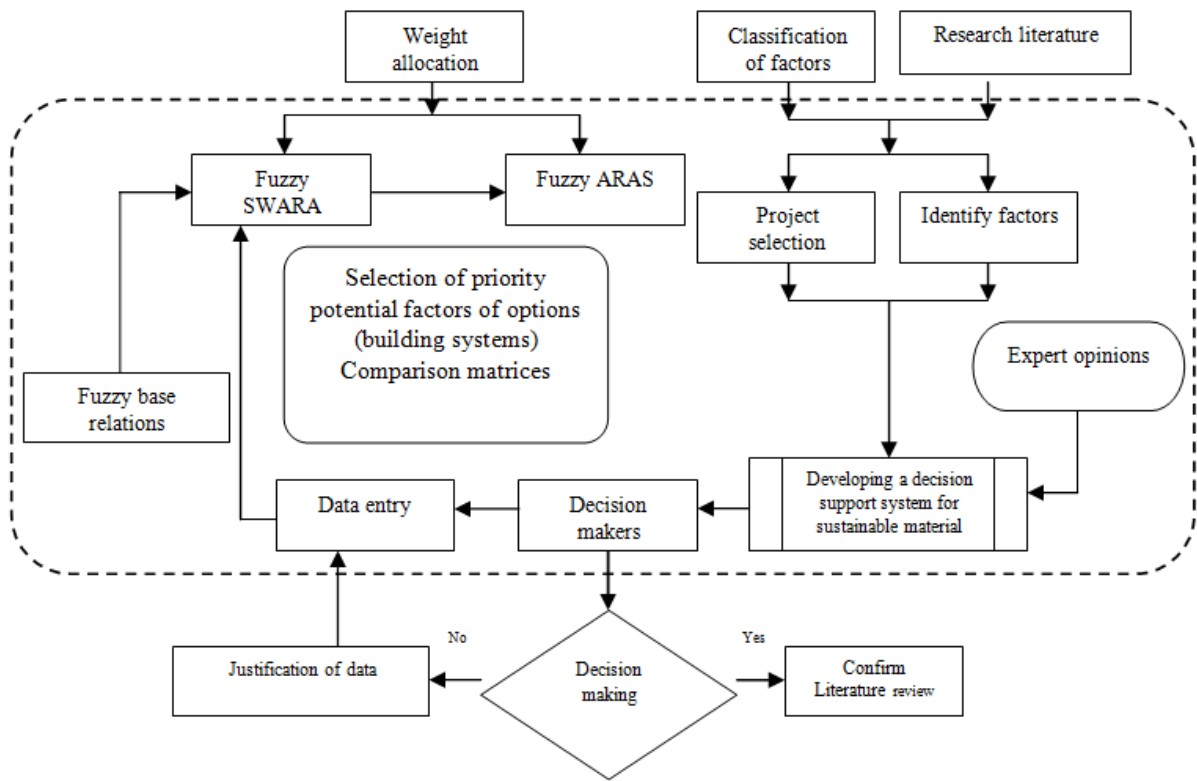

**Figure 5.** Structure of the hybrid fuzzy decision-making model.

In order to implement the aforementioned method, first, the questionnaires related to phase II were distributed among the experts, and they were asked to comparatively prioritize the efficient key factors using the expressions. In the next step, a workshop on the characteristics of each building system was organized, and the third-phase questionnaires were distributed among the experts. They were asked to comparatively prioritize building systems in terms of final efficient key factors using the expressions presented in Table 4. The SWARA (Stepwise Weight Assessment Ratio Analysis) method is an expert-centered method which was introduced by Keršuliene et al. [48]. It is an alternative to the AHP and ANP methods for prioritization analysis of decision-making issues. When compared to the older MCDM methods, this technique has a number of advantages as it requires a smaller number of comparisons; is more effective in terms of using the expert knowledge, experience, tacit information, and free evaluation of criteria without using scales; simple formulation; and time saving [49,50]. Furthermore, because the SWARA method ranks criteria in a descending order, there is no need to check for consistency in judgments [51]. A new additive ratio assessment (ARAS) method, which was introduced by Zavadskas and Turskis (2010) [52], selects the best option for a given strategic decision-making problem.

The performance of this method is comparable to that of the older methods such as the ideal solution (TOPSIS), VIKOR, and ELECTRE. It determines the ideal option by calculating the performance level of each option and its corresponding ratio [53,54].

Simple computational steps, separation of positive (benefit) and negative (cost) criteria, and evaluation and prioritization of different options based on a large number of criteria identified independently (without the need for many pairwise comparisons) are some of the primary advantages of the ARAS method over other decision-making methods [54,55]. This method is based on the idea that the phenomena of the complex world can be understood with reasonable accuracy using relatively simple comparisons [52].

The SWARA and ARAS methods, like the AHP method, are insufficient for determining parameter uncertainty, especially when explicit verbal expressions are used [56]. Due to the existing ambiguity concerning information in fuzzy numbers, Stanujkic [57] developed these methods. There are six main steps for the SWARA and ARAS fuzzy methods. Tables 5 and 6 present the details of the calculations required to implement the above mentioned methods, step by step.

**Table 5.** Stepwise description of decision-making using fuzzy SWARA method.

| Step | Description | Equations | |
|------|-------------|-----------|---|
| 1 | Sort the descending criteria in order of importance | In this step, the efficient factors extracted from the previous phase are arranged in descending order based on the fuzzy Delphi method. | |
| 2 | Determining the relative importance of fuzzy ($\widetilde{S}_j$) factor j compared to the previous factor ($j-1$) with more importance according to experts | In this step, the relative importance of each criterion compared to the previous criterion is determined using the verbal expressions in Table 3. | |
| 3 | Determination of fuzzy coefficient ($\widetilde{k}_j$) | $\widetilde{k}_j = \begin{cases} 1 & j = 1 \\ \widetilde{S}_j + 1 & j > 1 \end{cases}$ | (7) |
| 4 | Determining the initial fuzzy weight ($\widetilde{q}_j$) | $\widetilde{q}_j = \begin{cases} 1 & j = 1 \\ \widetilde{k}_{j-1}/\widetilde{k}_j & j > 1 \end{cases}$ | (8) |
| 5 | Determining the relative fuzzy weight of evaluation criteria ($\widetilde{w}_j$) | $\widetilde{w}_j = \widetilde{q}_j / \sum\limits_{k=1}^{n} \widetilde{q}_k$ | (9) |
| 6 | Defuzzification relative fuzzy weight of criterion j using region center method | $w_j = \frac{1}{3}\widetilde{w}_j =$ $\frac{1}{3}\left(\left(w_{j\gamma} - w_{j\alpha}\right) + \left(w_{j\beta} - w_{j\alpha}\right) + w_{j\alpha}\right)$ $w_{j\alpha}, w_{j\beta}$ and $w_{j\gamma}$ are the lower, middle, and upper bounds representing the relative fuzzy weights of the factors, respectively. | (10) |

**Table 6.** Stepwise description of decision-making using fuzzy ARAS method.

| Step | Description | Equations | |
|------|-------------|-----------|---|
| 1 | Fuzzy decision matrix formation: criterion-option matrix | $\widetilde{X} = \widetilde{x}_{ij} = \begin{bmatrix} \widetilde{x}_{11} & \cdots & \widetilde{x}_{1j} & \widetilde{x}_{0n} \\ \cdots & \cdots & \cdots & \cdots \\ \widetilde{x}_{i1} & \cdots & \widetilde{x}_{ij} & \widetilde{x}_{in} \\ \widetilde{x}_{m1} & \cdots & \widetilde{x}_{mj} & \widetilde{x}_{mn} \end{bmatrix} i = 0,1,\ldots,m; j = 1,2,\ldots,n$ $\widetilde{x}_{ij}$ is a fuzzy number that represents the performance of the *i*-th option in the *j*-th criterion. *m* is the number of options, and n is the number of criteria. | (11) |
| | | To form a fuzzy decision matrix, a row named the hypothetical ideal optimal value for the criteria ($\widetilde{x}_{0j}$ optimal value for the *j*-th criterion) is calculated as follows: $\widetilde{x}_{0j} = \underset{i}{Max}\,\widetilde{x}_{ij}$ if $\underset{i}{Max}\,\widetilde{x}_{ij}$, is prefreable $\widetilde{x}_{0j} = \underset{i}{Min}\,\widetilde{x}_{ij}$ if $\underset{i}{Min}\,\widetilde{x}_{ij}$, is prefreable Accordingly, positive criteria (such as profit: criteria whose increase improves the situation) with higher values and negative criteria (such as cost: criteria whose reduction is more economical) with lower values are preferred. | (12) |

**Table 6.** *Cont.*

| Step | Description | Equations | |
|------|-------------|-----------|---|
| 2 | Normalization of the decision matrix | $$\widetilde{\overline{X}} = \begin{bmatrix} \widetilde{\overline{x}}_{01} & \cdots & \widetilde{\overline{x}}_{0j} & \widetilde{\overline{x}}_{0n} \\ \cdots & \cdots & \cdots & \cdots \\ \widetilde{\overline{x}}_{i1} & \cdots & \widetilde{\overline{x}}_{ij} & \widetilde{\overline{x}}_{in} \\ \widetilde{\overline{x}}_{m1} & \cdots & \widetilde{\overline{x}}_{mj} & \widetilde{\overline{x}}_{mn} \end{bmatrix} \quad i = 0,1,\ldots,m; j = 1,2,\ldots,n$$ $\widetilde{\overline{x_{ij}}}$ are normalized values of matrix $\widetilde{\overline{X}}$ elements. | (13) |
| | | At this stage, the positive and negative criteria are normalized separately according to the following equations: $$\widetilde{\overline{x}}_{ij} = \widetilde{x}_{ij} / \sum_{i=0}^{m} \widetilde{x}_{ij}; \widetilde{\overline{x}}_{ij} = \frac{1}{\widetilde{\overline{x}}_{ij}}, \widetilde{\overline{x}}_{ij} = \widetilde{x}_{ij} / \sum_{i=0}^{m} \widetilde{x}_{ij}$$ | (14) |
| 3 | Formation of a normal balanced decision matrix | $$\widetilde{\widehat{X}} = \begin{bmatrix} \widetilde{\widehat{x}}_{01} & \cdots & \widetilde{\widehat{x}}_{0j} & \widetilde{\widehat{x}}_{0n} \\ \cdots & \cdots & \cdots & \cdots \\ \widetilde{\widehat{x}}_{i1} & \cdots & \widetilde{\widehat{x}}_{ij} & \widetilde{\widehat{x}}_{in} \\ \widetilde{\widehat{x}}_{m1} & \cdots & \widetilde{\widehat{x}}_{mj} & \widetilde{\widehat{x}}_{mn} \end{bmatrix} \quad i = 0,1,\ldots,m; j = 1,2,\ldots,n$$ $$\widetilde{\widehat{x}}_{ij} = w_j \widetilde{\overline{x}}_{ij} \quad i = 0,1,\ldots,m; j = 1,2,\ldots,n$$ It is sufficient to use the final weights obtained using methods such as Shannon entropy or FAHP to assign the initial weight to the criteria in the decision matrix. The weight obtained from the SWARA method is used as the initial weight of the criteria in this step. | (15) (16) |
| 4 | Determining the $\widetilde{S}_i$ value of the optimization function of the *i*-th option (the degree of utility of each option) | $$\widetilde{S}_i = \sum_{j=1}^{n} \widetilde{\widehat{x}}_{ij} \quad i = 0,1,\ldots,m; j = 1,2,\ldots,n$$ The higher the value $\widetilde{S}_i$, the better the option. | (17) |
| 5 | Defuzzification of optimization function with region center method (as the simplest method) | $$S_i = \frac{1}{3}\widetilde{S}_i = \frac{1}{3}\left( \left( S_{i\gamma} - S_{i\alpha} \right) + \left( S_{i\beta} - S_{i\alpha} \right) + S_{i\alpha} \right)$$ $S_{i\alpha}$, $S_{i\beta}$, and $S_{i\gamma}$ represent the most pessimistic, probable, and optimistic values of the TFNs. | (18) |
| 6 | Prioritization of options by calculating the degree of desirability | $$Q_i = S_i / S_0 \quad i = 0,1,\ldots,m$$ $Q_i$ is the degree of usefulness or relative degree of desirability of each option, and $S_0$ is the most desirable option. The higher the degree of desirability for an option, the better it will be prioritized and ranked. | (19) |

## 4. Findings

### 4.1. Results of Phase I: Identification and Classification of Factors

The results obtained from phase I as well as the FDM (1st and 2nd rounds) were used to create the database. This database consists of factors affecting the sustainable material selection, the main factor differentiation indicators, and the basic information on the five structural systems considered for execution of the building structural frames in the mass-housing projects. After final differentiation, a total number of 45 factors were classified based on the three main indicators of sustainable development (economic indicator ($C_2$), environmental indicator ($C_3$), socio-cultural indicator ($C_4$)), and an additional indicator known as technical-executive indicator ($C_1$). Since the present study was an attempt to apply the proposed support system to a real mass-housing project, the $C_1$ indicator was also taken into account in the study. The main indicators and subfactors associated with each indicator, as well as their codes and references, are listed in Table 7. The reliability and validity of the main indicators are presented in this table. Since $R_\alpha$ and $CVR$ values exceed the allowable limits considered for the main indicators, the designed questionnaire can be considered reliable and valid in terms of internal consistency of items.

**Table 7.** The effective main indicators and subfactors in the sustainable material selection process.

| Indicators (Code) (*CVR*, $R_\alpha$) | Factors (Code) | References |
|---|---|---|
| Technical-executive ($C_1$) (0.64, 0.84) | Manufacturability ($C_{1-1}$) | [6,16,18,20,27,31] |
| | Implementation ($C_{1-2}$) | [1,18,19,31] |
| | Repairability and maintainability ($C_{1-3}$) | [20,29] |
| | Easiness and speed in usability ($C_{1-4}$) | [21,30] |
| | Operational flexibility ($C_{1-5}$) | [3,6,14,17,21,26,30] |
| | Spatial scale ($C_{1-6}$) | [11,17,22,30,31] |
| | Demolition ($C_{1-7}$) | [12,17,20] |
| | Resistance to weathering, humidity, water, and fire ($C_{1-8}$) | [2,15,19,27] |
| | Hardness and weight savings ($C_{1-9}$) | [1,4,8,14,22,29] |
| | Compatibility with other material ($C_{1-10}$) | [4,13,18,24,28] |
| | Adaptation with technical standards ($C_{1-11}$) | [11,23,28] |
| | Resistance to erosion and corrosion ($C_{1-12}$) | [2,9,10,19,20,25,29] |
| | Durability ($C_{1-13}$) | [1,9,19,28] |
| | Expert labor ($C_{1-14}$) | [2,8–11,25,26,31] |
| Economics ($C_2$) (0.51, 0.89) | Material cost ($C_{2-1}$) | [1,6,23,24,30,31] |
| | Updated technology ($C_{2-2}$) | [1,15,24,28,30] |
| | Fabrication cost ($C_{2-3}$) | [3,5,8,16,17,23] |
| | Transportation cost ($C_{2-4}$) | [3,5,8,12,17,19] |
| | Life-cycle cost ($C_{2-5}$) | [7,13,18,24,29] |
| | Competitiveness cost ($C_{2-6}$) | [1,5,8,19,22,23,31] |
| | Repair and maintenance cost ($C_{2-7}$) | [1,3,14,15,23–25] |
| | Energy cost ($C_{2-8}$) | [1,14,15,24,25,30] |
| | Processing cost ($C_{2-9}$) | [1,12–14,21,23,28,29] |
| | Recycle cost ($C_{2-10}$) | [5,10,12,23,25] |
| Environmental ($C_3$) (0.62, 0.94) | Embodied energy ($C_{3-1}$) | [14,16,19,23,25] |
| | Acoustic resistance ($C_{3-2}$) | [12,15,17] |
| | Source material extraction ($C_{3-3}$) | [4,6,20,24,30] |
| | Energy consumption ($C_{3-4}$) | [2,3,8,26,31] |
| | Polluting ($C_{3-5}$) | [10,16,17,24,25] |
| | Environmental impacts ($C_{3-6}$) | [12,13,24–26,31] |
| | Reusability ($C_{3-7}$) | [9,19,24,29] |
| | Renewability ($C_{3-8}$) | [5,11,19,24,30] |
| | Compatibility with sustainable certifications ($C_{3-9}$) | [9,23,27] |
| | Disposal ($C_{3-10}$) | [1,15,25,30] |
| | Water savings ($C_{3-11}$) | [8,18,27–29] |
| | Climate change ($C_{3-12}$) | [14,15,22,29] |

**Table 7.** *Cont.*

| Indicators (Code) (CVR, $R_\alpha$) | Factors (Code) | References |
|---|---|---|
| Socio-cultural ($C_4$) (0.66, 0.88) | Human health and safety ($C_{4-1}$) | [11,14,17,19,20] |
| | Compatibility with ecology ($C_{4-2}$) | [3,13,22,29,31] |
| | Compatibility with identity ($C_{4-3}$) | [5,10,22,28] |
| | Flexibility about future plans ($C_{4-4}$) | [9,14,22,31] |
| | Use of local material ($C_{4-5}$) | [1,11,16,17,24,28,31] |
| | Productivity ($C_{4-6}$) | [1,7,17,19,21] |
| | Convenience ($C_{4-7}$) | [16,21,23,24,27,31] |
| | Human satisfaction ($C_{4-8}$) | [1,2,5,13,14,26,28,30] |
| | Aesthetic appeal ($C_{4-9}$) | [1,12,17,23] |

*4.2. Results of Phase II: Factor Monitoring and Screening*

In phase II, the key factors affecting the screening results, which were obtained by FDM (the 3rd round), were presented. The fuzzy and deterministic mean values for significance, class (efficient/inefficient), and type (cost/benefit) of each key factor are presented in Table 8. According to the results, 14 factors had deterministic mean values exceeding 0.5. These were identified as efficient factors in the selection of stable structural systems. In addition, the initial ranking of factors by the FDM showed that factors $C_{1-11}$, $C_{2-8}$, $C_{3-7}$, $C_{4-8}$, $C_{3-11}$, $C_{2-5}$, $C_{2-2}$, $C_{4-3}$, $C_{3-9}$, $C_{1-8}$, $C_{1-5}$, $C_{4-1}$, $C_{1-2}$, and $C_{3-4}$ were given higher comparative priority (respectively) in the process of sustainable material selection. In order to implement phase III, the extracted efficient factors were arranged in descending order and their type (benefit or cost) was specified.

*4.3. Results of Phase III: Prioritization of Factors and Selection of Alternatives*

The results obtained in phase III were divided into two sections. In the first section, the efficient factors extracted from phase II were prioritized to identify the final factors affecting the sustainable material selection. In this section, prior to starting the fuzzy SWARA calculations, 14 identified critical factors were ranked in terms of significance and in descending order based on the scores obtained from the FDM. In the next step, the relative fuzzy significance ($\widetilde{S}_j$) computed based on the expressions chosen by the experts, as well as fuzzy coefficient ($\widetilde{k}_j$), initial fuzzy weight ($\widetilde{q}_j$) relative fuzzy weight ($\widetilde{w}_j$), and defuzzification values obtained from the center of area (CoA) method, were used to determine the final key factors and their ranking. Computational details of the fuzzy SWARA method are presented in Table 9.

Table 8. Fuzzy mean, deterministic number, class, and type of factors affecting the sustainable material selection.

| Indicators | Key Factors | Fuzzy Average Comments | | | Defuzzification | | | Crisp Number | Type of Factors | Invoice Type (Cost/Benefit) | Descending Rank | |
|---|---|---|---|---|---|---|---|---|---|---|---|---|
| | | $\bar{a}^{l}$ | $\bar{a}^{m}$ | $\bar{a}^{u}$ | $x^{l}_{max}$ | $x^{m}_{max}$ | $x^{u}_{max}$ | | | | Group | General |
| $C_1$ | $C_{1-1}$ | 0.229 | 0.417 | 0.667 | 0.438 | 0.427 | 0.438 | 0.438 | Inefficient | − | 9 | 28 |
| | $C_{1-2}$ | 0.271 | 0.521 | 0.771 | 0.521 | 0.521 | 0.516 | 0.521 | Efficient | Benefit | 4 | 13 |
| | $C_{1-3}$ | 0.271 | 0.396 | 0.583 | 0.417 | 0.406 | 0.391 | 0.417 | Inefficient | − | 12 | 33 |
| | $C_{1-4}$ | 0.146 | 0.229 | 0.479 | 0.285 | 0.257 | 0.297 | 0.297 | Inefficient | − | 14 | 45 |
| | $C_{1-5}$ | 0.292 | 0.542 | 0.792 | 0.542 | 0.542 | 0.531 | 0.542 | Efficient | Cost | 3 | 11 |
| | $C_{1-6}$ | 0.104 | 0.292 | 0.542 | 0.313 | 0.302 | 0.344 | 0.344 | Inefficient | − | 13 | 43 |
| | $C_{1-7}$ | 0.229 | 0.458 | 0.667 | 0.451 | 0.455 | 0.448 | 0.455 | Inefficient | − | 7 | 25 |
| | $C_{1-8}$ | 0.375 | 0.563 | 0.729 | 0.556 | 0.559 | 0.505 | 0.559 | Efficient | Cost | 2 | 10 |
| | $C_{1-9}$ | 0.229 | 0.417 | 0.625 | 0.424 | 0.420 | 0.417 | 0.424 | Inefficient | − | 11 | 32 |
| | $C_{1-10}$ | 0.292 | 0.500 | 0.688 | 0.493 | 0.497 | 0.469 | 0.497 | Inefficient | − | 5 | 15 |
| | $C_{1-11}$ | 0.417 | 0.667 | 0.854 | 0.646 | 0.656 | 0.594 | 0.656 | Efficient | Benefit | 1 | 1 |
| | $C_{1-12}$ | 0.208 | 0.438 | 0.688 | 0.444 | 0.441 | 0.453 | 0.453 | Inefficient | − | 8 | 27 |
| | $C_{1-13}$ | 0.250 | 0.500 | 0.708 | 0.486 | 0.493 | 0.479 | 0.493 | Inefficient | − | 6 | 16 |
| | $C_{1-14}$ | 0.292 | 0.417 | 0.583 | 0.431 | 0.424 | 0.396 | 0.431 | Inefficient | − | 10 | 29 |
| $C_2$ | $C_{2-1}$ | 0.208 | 0.354 | 0.563 | 0.375 | 0.365 | 0.370 | 0.375 | Inefficient | − | 13 | 42 |
| | $C_{2-2}$ | 0.375 | 0.583 | 0.792 | 0.583 | 0.583 | 0.542 | 0.583 | Efficient | Cost | 3 | 7 |
| | $C_{2-3}$ | 0.146 | 0.354 | 0.604 | 0.368 | 0.361 | 0.391 | 0.391 | Inefficient | − | 12 | 39 |
| | $C_{2-4}$ | 0.250 | 0.375 | 0.604 | 0.410 | 0.392 | 0.396 | 0.410 | Inefficient | − | 10 | 36 |
| | $C_{2-5}$ | 0.354 | 0.604 | 0.792 | 0.583 | 0.594 | 0.547 | 0.594 | Efficient | Benefit | 2 | 6 |
| | $C_{2-6}$ | 0.208 | 0.375 | 0.604 | 0.396 | 0.385 | 0.396 | 0.396 | Inefficient | − | 11 | 38 |
| | $C_{2-7}$ | 0.271 | 0.479 | 0.646 | 0.465 | 0.472 | 0.443 | 0.472 | Inefficient | − | 6 | 20 |
| | $C_{2-8}$ | 0.417 | 0.646 | 0.833 | 0.632 | 0.639 | 0.578 | 0.639 | Efficient | Cost | 1 | 2 |
| | $C_{2-9}$ | 0.229 | 0.458 | 0.667 | 0.451 | 0.455 | 0.448 | 0.455 | Inefficient | − | 7 | 25 |
| | $C_{2-10}$ | 0.292 | 0.479 | 0.688 | 0.486 | 0.483 | 0.464 | 0.486 | Inefficient | − | 5 | 17 |
| $C_3$ | $C_{3-1}$ | 0.229 | 0.458 | 0.708 | 0.465 | 0.462 | 0.469 | 0.469 | Inefficient | − | 5 | 21 |
| | $C_{3-2}$ | 0.208 | 0.417 | 0.646 | 0.424 | 0.420 | 0.427 | 0.427 | Inefficient | − | 9 | 31 |
| | $C_{3-3}$ | 0.250 | 0.458 | 0.667 | 0.458 | 0.458 | 0.448 | 0.458 | Inefficient | − | 7 | 23 |
| | $C_{3-4}$ | 0.271 | 0.521 | 0.729 | 0.507 | 0.514 | 0.495 | 0.514 | Efficient | Benefit | 4 | 14 |
| | $C_{3-5}$ | 0.167 | 0.396 | 0.625 | 0.396 | 0.396 | 0.411 | 0.411 | Inefficient | − | 10 | 34 |
| | $C_{3-6}$ | 0.188 | 0.396 | 0.625 | 0.403 | 0.399 | 0.411 | 0.411 | Inefficient | − | 10 | 34 |
| | $C_{3-7}$ | 0.396 | 0.625 | 0.854 | 0.625 | 0.625 | 0.583 | 0.625 | Efficient | Benefit | 1 | 3 |
| | $C_{3-8}$ | 0.250 | 0.458 | 0.667 | 0.458 | 0.458 | 0.448 | 0.458 | Inefficient | − | 7 | 23 |
| | $C_{3-9}$ | 0.333 | 0.563 | 0.813 | 0.569 | 0.566 | 0.547 | 0.569 | Efficient | Cost | 3 | 9 |
| | $C_{3-10}$ | 0.292 | 0.458 | 0.646 | 0.465 | 0.462 | 0.438 | 0.465 | Inefficient | − | 6 | 22 |
| | $C_{3-11}$ | 0.375 | 0.604 | 0.792 | 0.590 | 0.597 | 0.547 | 0.597 | Efficient | Cost | 2 | 5 |
| | $C_{3-12}$ | 0.208 | 0.375 | 0.563 | 0.382 | 0.378 | 0.375 | 0.382 | Inefficient | − | 12 | 41 |

**Table 8.** *Cont.*

| Indicators | Key Factors | Fuzzy Average Comments | | | Defuzzification | | | Crisp Number | Type of Factors | Invoice Type (Cost/Benefit) | Descending Rank | |
| | | $\overset{-l}{a}$ | $\overset{-m}{a}$ | $\overset{-u}{a}$ | $x^l_{max}$ | $x^m_{max}$ | $x^u_{max}$ | | | | Group | General |
|---|---|---|---|---|---|---|---|---|---|---|---|---|
| | $C_{4-1}$ | 0.313 | 0.542 | 0.750 | 0.535 | 0.538 | 0.510 | 0.538 | Efficient | Benefit | 3 | 12 |
| | $C_{4-2}$ | 0.104 | 0.271 | 0.521 | 0.299 | 0.285 | 0.328 | 0.328 | Inefficient | – | 9 | 44 |
| | $C_{4-3}$ | 0.375 | 0.583 | 0.771 | 0.576 | 0.580 | 0.531 | 0.580 | Efficient | Cost | 2 | 8 |
| | $C_{4-4}$ | 0.271 | 0.417 | 0.604 | 0.431 | 0.424 | 0.406 | 0.431 | Inefficient | – | 6 | 30 |
| $C_4$ | $C_{4-5}$ | 0.167 | 0.354 | 0.604 | 0.375 | 0.365 | 0.391 | 0.391 | Inefficient | – | 8 | 39 |
| | $C_{4-6}$ | 0.188 | 0.375 | 0.625 | 0.396 | 0.385 | 0.406 | 0.406 | Inefficient | – | 7 | 37 |
| | $C_{4-7}$ | 0.292 | 0.458 | 0.688 | 0.479 | 0.469 | 0.458 | 0.479 | Inefficient | – | 4 | 18 |
| | $C_{4-8}$ | 0.354 | 0.604 | 0.833 | 0.597 | 0.601 | 0.568 | 0.601 | Efficient | Benefit | 1 | 4 |
| | $C_{4-9}$ | 0.292 | 0.479 | 0.667 | 0.479 | 0.479 | 0.453 | 0.479 | Inefficient | – | 4 | 18 |

**Table 9.** Fuzzy SWARA computation results used to determine the ranking of efficient key factors.

| Indicators | Key Factors | $\tilde{s}_j$ | | | $\tilde{k}_j$ | | | $\tilde{q}_j$ | | | $\tilde{w}_j$ | | | $w_j$ | | Rank | |
| | | $s_{j\alpha}$ | $s_{j\beta}$ | $s_{j\gamma}$ | $k_{j\alpha}$ | $k_{j\beta}$ | $k_{j\gamma}$ | $q_{j\alpha}$ | $q_{j\beta}$ | $q_{j\gamma}$ | $w_{j\alpha}$ | $w_{j\beta}$ | $w_{j\gamma}$ | Explicit | Normalized | Initial | Final |
|---|---|---|---|---|---|---|---|---|---|---|---|---|---|---|---|---|---|
| | $C_{1-11}$ | 0 | 0 | 0 | 1 | 1 | 1 | 1 | 1 | 1 | 0.188 | 0.202 | 0.220 | 0.0565 | 0.2494 | 2 | 11 |
| $C_1$ | $C_{1-8}$ | 0.250 | 0.438 | 0.646 | 1.250 | 1.438 | 1.646 | 1.250 | 1.438 | 1.646 | 0.234 | 0.290 | 0.362 | 0.0542 | 0.2393 | 4 | 13 |
| | $C_{1-5}$ | 0.273 | 0.521 | 0.682 | 1.273 | 1.521 | 1.682 | 1.018 | 1.058 | 1.022 | 0.191 | 0.214 | 0.225 | 0.0599 | 0.2646 | 1 | 8 |
| | $C_{1-2}$ | 0.300 | 0.542 | 0.700 | 1.300 | 1.542 | 1.700 | 1.277 | 1.457 | 1.664 | 0.239 | 0.294 | 0.366 | 0.0559 | 0.2468 | 3 | 12 |
| | $C_{2-8}$ | 0 | 0 | 0 | 1 | 1 | 1 | 1 | 1 | 1 | 0.269 | 0.293 | 0.304 | 0.0863 | 0.3412 | 2 | 3 |
| $C_2$ | $C_{2-5}$ | 0.271 | 0.479 | 0.688 | 1.271 | 1.479 | 1.688 | 1.271 | 1.479 | 1.688 | 0.342 | 0.434 | 0.51 | 0.0878 | 0.3473 | 1 | 1 |
| | $C_{2-2}$ | 0.295 | 0.375 | 0.727 | 1.295 | 1.375 | 1.727 | 1.019 | 0.930 | 1.024 | 0.275 | 0.273 | 0.31 | 0.0787 | 0.3115 | 3 | 6 |
| | $C_{3-7}$ | 0 | 0 | 0 | 1 | 1 | 1 | 1 | 1 | 1 | 0.185 | 0.202 | 0.22 | 0.0553 | 0.2605 | 2 | 9 |
| $C_3$ | $C_{3-11}$ | 0.292 | 0.521 | 0.729 | 1.292 | 1.521 | 1.729 | 1.292 | 1.521 | 1.729 | 0.239 | 0.308 | 0.383 | 0.0546 | 0.2574 | 3 | 10 |
| | $C_{3-9}$ | 0.273 | 0.604 | 0.705 | 1.273 | 1.604 | 1.705 | 0.985 | 1.055 | 0.986 | 0.182 | 0.214 | 0.218 | 0.0591 | 0.2786 | 1 | 7 |
| | $C_{3-4}$ | 0.225 | 0.438 | 0.675 | 1.225 | 1.438 | 1.675 | 1.243 | 1.363 | 1.699 | 0.230 | 0.276 | 0.376 | 0.0432 | 0.2035 | 4 | 14 |
| | $C_{4-8}$ | 0 | 0 | 0 | 1 | 1 | 1 | 1 | 1 | 1 | 0.269 | 0.280 | 0.305 | 0.0812 | 0.3256 | 3 | 5 |
| $C_4$ | $C_{4-3}$ | 0.292 | 0.521 | 0.729 | 1.292 | 1.521 | 1.729 | 1.292 | 1.521 | 1.729 | 0.348 | 0.425 | 0.528 | 0.0818 | 0.3278 | 2 | 4 |
| | $C_{4-1}$ | 0.273 | 0.604 | 0.705 | 1.273 | 1.604 | 1.705 | 0.985 | 1.055 | 0.986 | 0.265 | 0.295 | 0.301 | 0.0865 | 0.3466 | 1 | 2 |

Based on the results of this section, the researchers managed to select the three high–ranking factors (final factors) associated with each of the main indicators. According to the results, operational flexibility ($C_{1-5}$), life-cycle cost ($C_{2-5}$), compatibility with sustainable certifications ($C_{3-9}$), and human health and safety ($C_{4-1}$) are ranked as first by $C_1$, $C_2$, $C_3$, and $C_4$ indicators, respectively. The results also show that, adaptation with technical standards ($C_{1-11}$), energy cost ($C_{2-8}$), reusability ($C_{3-7}$), and compatibility with identity ($C_{4-3}$) are ranked as second by each of the mentioned indicators, respectively.

In the second section of this phase, the Fuzzy ARAS method was used to rank the structural systems in terms of accessibility of factors affecting the sustainable material selection in mass-housing projects. In the structural system ranking process, the fuzzy ARAS method was used to determine the frequency of key factors and the fuzzy decision-making matrix corresponding to each factor based on the opinions of experts. This matrix consisted of 17 final factors (criteria) and 5 building systems (alternatives). The computational process as well as the fuzzy decision-making matrix development process for the different structural factors and systems are briefly presented in Table 10. The first row of this matrix encompasses the hypothetical ideal value for the criteria (A0). The weight obtained from the fuzzy SWARA method (the method used to rank factors in the previous section) was taken as the initial weight of the criteria in the decision matrix. The benefit–cost criteria were also specified for each of the alternatives. Once the fuzzy decision matrix was normalized and a weighted normalized matrix was derived from it, the utility (degree of utility) fuzzy function ($\widetilde{S}_i$) was calculated (Tables 10 and 11) with respect to $S_{i\alpha}$, $S_{i\beta}$, and $S_{i\gamma}$ which represent the most pessimistic, probable, and optimistic values of the triangular fuzzy number, respectively. Finally, the obtained value was defuzzified using the COD method in order to obtain the relative degree of utility ($Q_i$) of each alternative (structural system) with respect to each of the key factors. The final rank of the alternatives was determined as shown in Figure 6. The results could be used to evaluate different alternatives in terms of key factors affecting the sustainable material selection. For instance, in this case, the comparative priority of the alternatives was examined based on the most important key factors obtained from the previous phase ($C_{1-5}$, $C_{2-5}$, $C_{3-9}$, and $C_{4-1}$). The results showed that PRC, ICF, and LSF systems had respectively higher comparative priorities in terms of factor $C_{1-5}$. However, when it came to factor $C_{2-5}$, LSF, PRC, and ICF systems were more preferable, respectively. PRC, ICF, and 3DP systems were found to be among the top ranking priorities in terms of factor $C_{3-9}$. LSF, ICF, and PRC systems were found to be most desirable (respectively) with respect to factor $C_{4-1}$. Finally, the overall alternative ranking results showed that LSF, ICF, and PRC systems with utility degrees of 1.800, 1.614, and 1.536, could be identified as the most preferred systems in terms of meeting sustainable material selection goals in mass-housing projects.

**Table 10.** Summary of the fuzzy ARAS computations and decision-matrix development process for each key factor and structural system.

| Alternatives | Final Key Factors | | | | | | | | | | | |
|---|---|---|---|---|---|---|---|---|---|---|---|---|
| | Decision Matrix | | | | | | | | | | | |
| | $C_{1-11}$ | | | $C_{1-8}$ | | | $C_{1-5}$ | | | $C_{4-1}$ | | |
| A0 | 0.188 | 0.202 | 0.22 | 0.234 | 0.29 | 0.362 | 0.191 | 0.214 | 0.225 | 0.265 | 0.295 | 0.301 |
| LSF | 0.208 | 0.229 | 0.250 | 0.389 | 0.417 | 0.445 | 0.611 | 0.667 | 0.723 | 0.455 | 0.514 | 0.573 |
| PRC | 0.458 | 0.479 | 0.500 | 0.701 | 0.729 | 0.757 | 0.882 | 0.938 | 0.994 | 0.521 | 0.580 | 0.639 |
| ICF | 0.542 | 0.563 | 0.584 | 0.743 | 0.771 | 0.799 | 0.798 | 0.854 | 0.910 | 0.424 | 0.483 | 0.542 |
| 3DP | 0.167 | 0.188 | 0.209 | 0.243 | 0.271 | 0.299 | 0.423 | 0.479 | 0.535 | 0.399 | 0.458 | 0.517 |
| TRC | 0.083 | 0.104 | 0.125 | 0.264 | 0.292 | 0.320 | 0.465 | 0.521 | 0.577 | 0.399 | 0.458 | 0.517 |
| | Normalized decision matrix | | | | | | | | | | | |
| A0 | 0.099 | 0.114 | 0.133 | 0.078 | 0.104 | 0.140 | 0.048 | 0.058 | 0.066 | 0.085 | 0.105 | 0.122 |
| LSF | 0.110 | 0.129 | 0.152 | 0.130 | 0.150 | 0.172 | 0.154 | 0.181 | 0.214 | 0.147 | 0.184 | 0.232 |
| PRC | 0.242 | 0.271 | 0.304 | 0.235 | 0.263 | 0.294 | 0.222 | 0.255 | 0.295 | 0.168 | 0.208 | 0.259 |

**Table 10.** *Cont.*

| Alternatives | Final Key Factors | | | | | | | | | | | |
|---|---|---|---|---|---|---|---|---|---|---|---|---|
| | Decision Matrix | | | | | | | | | | | |
| | $C_{1-11}$ | | | $C_{1-8}$ | | | $C_{1-5}$ | | | $C_{4-1}$ | | |
| | Normalized decision matrix | | | | | | | | | | | |
| ICF | 0.287 | 0.319 | 0.354 | 0.249 | 0.278 | 0.310 | 0.201 | 0.232 | 0.270 | 0.137 | 0.173 | 0.22 |
| 3DP | 0.088 | 0.106 | 0.126 | 0.081 | 0.098 | 0.116 | 0.107 | 0.130 | 0.159 | 0.129 | 0.164 | 0.210 |
| TRC | 0.044 | 0.059 | 0.076 | 0.088 | 0.105 | 0.124 | 0.117 | 0.142 | 0.171 | 0.129 | 0.164 | 0.210 |
| W | 0.056 | 0.056 | 0.056 | 0.054 | 0.054 | 0.054 | 0.06 | 0.056 | 0.056 | 0.086 | 0.086 | 0.086 |
| | Weighted normalized decision matrix | | | | | | | | | | | |
| A0 | 0.005 | 0.006 | 0.007 | 0.004 | 0.005 | 0.007 | 0.003 | 0.003 | 0.004 | 0.007 | 0.009 | 0.010 |
| LSF | 0.006 | 0.007 | 0.008 | 0.007 | 0.008 | 0.009 | 0.009 | 0.011 | 0.013 | 0.012 | 0.016 | 0.020 |
| PRC | 0.013 | 0.015 | 0.017 | 0.012 | 0.014 | 0.016 | 0.013 | 0.015 | 0.017 | 0.014 | 0.018 | 0.022 |
| ICF | 0.016 | 0.018 | 0.020 | 0.013 | 0.015 | 0.017 | 0.012 | 0.014 | 0.016 | 0.012 | 0.015 | 0.019 |
| 3DP | 0.005 | 0.006 | 0.007 | 0.004 | 0.005 | 0.006 | 0.006 | 0.007 | 0.009 | 0.011 | 0.014 | 0.018 |
| TRC | 0.002 | 0.003 | 0.004 | 0.005 | 0.005 | 0.006 | 0.007 | 0.008 | 0.010 | 0.011 | 0.014 | 0.018 |

**Table 11.** Results of structural system ranking based on utility function (degree of desirability) in fuzzy ARAS method.

| Alternatives | $\tilde{S}_i$ | | | Crisp $S_i$ | $Q_i$ | Final Rank |
|---|---|---|---|---|---|---|
| | $S_{i\alpha}$ | $S_{i\beta}$ | $S_{i\gamma}$ | | | |
| A0 | 0.08336 | 0.10544 | 0.13607 | 0.0527 | – | – |
| LSF | 0.17465 | 0.20860 | 0.25072 | 0.0949 | 1.800 | 1 |
| PRC | 0.14132 | 0.17261 | 0.21163 | 0.0810 | 1.536 | 3 |
| ICF | 0.15343 | 0.18485 | 0.22378 | 0.0851 | 1.614 | 2 |
| 3DP | 0.10394 | 0.13119 | 0.16529 | 0.0642 | 1.217 | 5 |
| TRC | 0.11049 | 0.13857 | 0.17370 | 0.0673 | 1.276 | 4 |

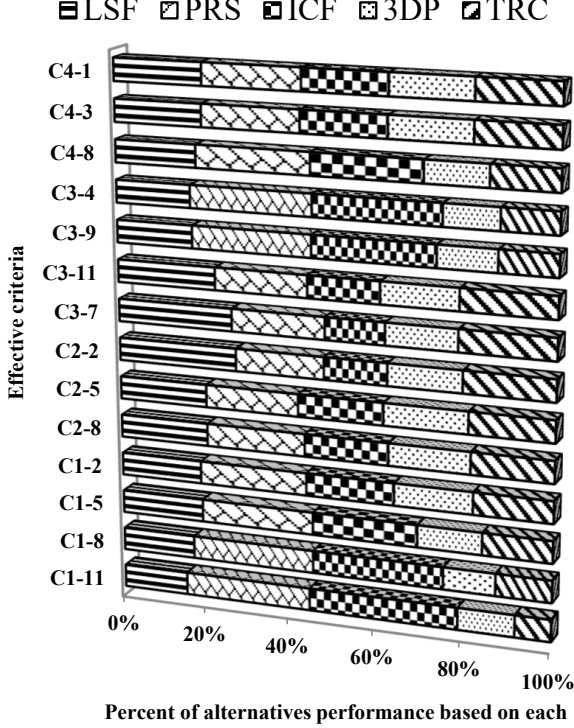

**Figure 6.** Degree of utility and ranking of the structural systems according to each of the key factors.

## 5. Conclusions

The construction industry in Iran is recognized as one of the most important industries involving the national economy and people's livelihoods. It is a long-term and complex task of implementing sustainable development in Iran, where opportunities and challenges coexist. Since selection of the sustainable materials and structural systems involves many affecting factors, identifying the core drivers of this selection can help governments, industry organizations, and enterprise management achieve the sustainable development goals faster and better.

In the present paper, a decision-making support model was proposed for evaluating the factors affecting the selection of sustainable materials, as well as choosing the optimal structural system in building projects. The methodology of the proposed decision-making support model consists of a three-phase protocol with integrated Delphi–SWARA–ARAS methods in a fuzzy environment.

In phase I, an in-depth review of the literature was used to identify the most important factors affecting the sustainable materials selection in construction projects. These drivers included 42 main factors. At the next step, the identified factors were assigned to groups of main indicators, and the content validity of the indicators was determined based on the experts' opinions. In phase II, Likert scale-based structured questionnaires were designed to elicit the experts' opinions on the significance of the factors. Afterwards, taking into account the uncertainties associated with the opinions of experts, the researchers used the fuzzy Delphi method (FDM) to determine the relative significance of the factors. In phase III, taking into account the uncertainties related to the decision-making process, the researchers used hybrid fuzzy SWARA–fuzzy ARAS methods to screen and prioritize the efficient key factors in the process of sustainable material selection. Finally, the preferred structural systems for mass-housing projects were selected. The 42 identified factors in phase I, using the FDM, were classified into four main indicators (technical-executive ($C_1$), economic ($C_2$), environmental ($C_3$), socio-cultural ($C_4$)) in accordance with sustainable development aspects. The results of the fuzzy SWARA method were used to determine the key factors affecting the sustainable material selection. Based on the FDM implementation in phase II, the 14 identified factors were extracted as sufficient factors. These factors include: implementation ($C_{1-2}$); operational flexibility ($C_{1-5}$); resistance to weathering, humidity, water, and fire ($C_{1-8}$); adaptation with technical standards ($C_{1-11}$); updated technology ($C_{2-2}$); life-cycle cost ($C_{2-5}$); energy cost ($C_{2-8}$); energy consumption ($C_{3-4}$); reusability ($C_{3-7}$); compatibility with sustainable certifications ($C_{3-9}$); water savings ($C_{3-11}$); human health and safety ($C_{4-1}$); compatibility with identity ($C_{4-3}$); and human satisfaction ($C_{4-8}$). The results showed that operational flexibility, life-cycle cost, compatibility with sustainable certifications, and human health and safety were the most important four factors affecting to sustainable material selection in the related $C_1$, $C_2$, $C_3$, and $C_4$ indicators. The results also showed that adaptation with technical standards, energy cost, reusability, and compatibility with identity were ranked as second by each of the mentioned indicators, respectively. The results of phase II based on the fuzzy SWARA method implementation were used to determine the key factors affecting the sustainable material selection. The results of this phase indicated that the factors associated with each main indicator can be comparatively prioritized as follows: indicator $C_1$ ($C_{1-4} > C_{1-2} > C_{1-5}$), indicator $C_2$ ($C_{2-5} > C_{2-3} > C_{2-1}$), indicator $C_3$ ($C_{3-8} > C_{3-4} > C_{3-6}$), and indicator $C_4$ ($C_{4-4} > C_{4-10} > C_{4-5}$). Finally, the results of the fuzzy ARAS method in phase III (during the process of ranking optimal structural systems based on 14 efficient key factors) indicated that the LSF, ICF, and PRC systems can be prioritized as the most preferred systems, respectively (in terms of fulfillment of sustainable development goals) in mass-housing projects. In other words, the aforementioned systems can play a more effective role (as compared to the identified subcriteria) in the implementation of the new building system in terms of sustainable material selection.

The proposed support model in the present paper can be used as a valuable benchmark by mass-housing construction beneficiaries seeking to increase the sustainability of

construction projects. The framework solved the problem that experts cannot give accurate quantitative judgments on real and complicated problems and used the fuzzy hybrid approaches and triangular fuzzy numbers in real-life methods. The analysis indicated that the adopted hybrid approach had good stability and robustness. Under different mutual weights, the sorting trend was basically the same. This paper sheds light on the key factors of sustainable material selection and the optimal structural system in mass-housing projects, filling the gap between the existing literature and sustainable material selection research. The analysis provides employer, contractor, and consultant companies at all levels, enterprise managers, engineers, technicians, and related researchers with a deeper understanding of sustainable material selection and its implementation to choose the optimal structural system in mass-housing projects.

Differences in research results may exist when this study method is applied to other regions of Iran. Although our proposed framework is a pioneering study using new hybrid methods in fuzzy environment on the implementation of sustainable material selection and optimal structural system, the empirical analysis and the fuzzy MCDM approach used in this research can still be used for evaluating the implementation of sustainable material selection in other countries. Additionally, the main difference and innovation of this research is the consideration of a large number of material selection criteria, and the presentation of a hybrid system with decision-making methods which have been used less in past research on this issue is one of the other innovations of this article. However, the approach proposed in the present study can utilize a larger number of criteria and alternatives to evaluate other construction projects, especially in cases where decision-making goals are contradictory.

**Author Contributions:** Conceptualization, E.A. and H.Y.; methodology, E.A.; software, E.A.; validation, E.A.; formal analysis, E.A.; investigation, E.A. and H.Y.; data curation, E.A. and H.Y.; writing—original draft preparation, H.Y. and M.K.; writing—review and editing, E.A., H.Y. and M.K.; visualization, E.A.; supervision, H.Y.; project administration, E.A., H.Y. and M.K. All authors have read and agreed to the published version of the manuscript.

**Funding:** This research received no external funding.

**Institutional Review Board Statement:** Not applicable.

**Informed Consent Statement:** Not applicable.

**Data Availability Statement:** Data available on request due to restrictions.

**Acknowledgments:** The authors are grateful for the valuable comments and suggestions of the respected reviewers. These comments enhanced the strength and significance of our paper.

**Conflicts of Interest:** Authors declare no conflict of interest.

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
