# Peer review of "A Hybrid Fuzzy MCDM Methodology for Optimal Structural System Selection Compatible with Sustainable Materials in Mass-Housing Projects"

_sustainability, doi:10.3390/su142013559_

Round 1

Reviewer 1 Report

Comments to Authors

In this manuscript, authors discuss an integrated fuzzy multi-criteria-decision-making model for the selection of sustainable materials in mass-housing projects. The proposed support system is implemented in order to apply a realistic scenario to a problem related to the selection of structural system for a mass-housing project in a residential area within Iran. In this way, an executing company selects the building materials in cooperation with the employer in order to make sure about compliance of construction process with the principles of sustainable development at decision-making stage.

Authors should address following comments before the decision on this manuscript can be made:

1.      The writing and organization of manuscript should be improved.

2.      It contains a few technical and various English language mistakes, which should be corrected.

3.      Various statements need some reliable citations.

4.      Authors should add some research contents which can improve the worth of research progress of this work.

5.      Conclusions section is too lengthy. It should be comprehensive and should reflect the outcomes of performed studies.

Reviewer 2 Report

In the present paper, analytical methods are used to propose a decision–making support model for evaluating the factors affecting the selection of sustainable materials and the optimal structural system in building projects. To this end, an integrated fuzzy multi-criteria–decision–making method is used to identify the criteria affecting the sustainable material selection.
The topic is essential and is the subject of research by many authors, which the authors themselves state.
The manuscript is difficult to follow. Many facts are repeated, especially in the Introductory part, which has been expanded. Or Conclusions from which it is challenging to understand the most important conclusions of the paper.
Significant correction of spelling and grammatical errors is necessary. There are many technical errors, and Author must check reference numbering. In part References: reference 1 and reference 2 are not listed! You should check that the numbering of other references is correct.
I suggest a major revision.
Document type: Review

Reviewer 3 Report

-        The title deals with the selection of materials but the content of the study raises the selection of composite structural systems. It is recommended to specify the title.

-        In the abstract section, the indication of the acronyms LSF, ICF and PRC, is not understood if it is not explained.

-        It is not adequately expressed what the objectives of the study are.

-        The selection of materials with sustainability criteria cannot be a statistical issue exclusively and must be based on technical and scientific data developed by manufacturers through EPDs (Environmental Product Declarations), where environmental impact data can be collected from materials and products throughout their Life Cycle Assessment (LCA).

-        Additionally, this assessment can be very relative and must be valued by the designer (architect, engineer, etc.) in the design phase, for decision making, rather than in the construction phases. Hence, table 2 is irrelevant or unappropriated as a sample of the study.

-        There is erratum in references 1 and 2. Review of specialized literature on LCA and EPD is recommended

-        In Table 1, reference the authors according to the references at the end of the paper. Include separations between the sections of the table. The same, in Table 7.

-        Name the number of phases in the same way in the text and tables: Phase II or Phase 2

-        Complete Table 3 with graphical documentation. Too many details are given about these products that are not decisive in the study.

-        The correspondence between results and conclusions is not observed, nor the true usefulness of the method as a tool in the selection of sustainable materials. Thanks

T

Reviewer 4 Report

This paper presents an optimal structural selection approach using an integrated fuzzy multi-criteria-decision making method through three sustainable development indicators. Though the paper presented an interesting approach, the authors need to address the following comments and revise the manuscript.

1.      The abstract should be rewritten to include the methodology, and assumptions and also expand LSF, ICF, and PRC before abbreviating.

2.      The authors should distinguish the contribution and novelty of this work from that of the existing similar works. Add or expand the last paragraph of the introduction to clearly state what is new in this paper.

3.      It looks like from the three steps used in the proposed decision-making approach, the paper heavily relies on and focused on the opinions of the 12- member experts. Justify this methodology?

4.      Only, LSF, ICF and PRC structural systems were studied. However, due to increasing improvements in manufacturing technology and material science, Functionally graded materials (FGMs) are gaining a huge interest in construction, particularly for extreme loading environments (fire and blast). Add a few pieces of literature to the introduction section such as:

·        Fire Safety Journal2021, 125, 103425

·        Buildings2022, 12(2), 118

5.      There is no Life cycle assessment (LCA) presented for each structural system to rate their eco-friendliness. Please clarify.

Round 2

Reviewer 1 Report

Comments to Authors

Authors have addressed a few of my comments. Anyhow, the manuscript can be accepted for publication in Sustainability.

Reviewer 2 Report

I propose that the revised manuscript be accepted.

Reviewer 4 Report

I don't see any significant improvement.  Most comments were not addressed and not clearly shown in the revised manuscript. Abstract, Introduction and conclusion sections are almost exactly same.

Please show clearly all the changes made based on the comments from the reviewer in the revised manuscript instead of highlighting the whole section!!

Round 3

Reviewer 4 Report

Thank you for revising the manuscript.